# Characterization of Three-Dimensional Trophoblast Spheroids: An Alternative Model to Study the Physiological Properties of the Placental Unit

**DOI:** 10.3390/cells11182884

**Published:** 2022-09-15

**Authors:** Violeta Stojanovska, Susanne Arnold, Mario Bauer, Hermann Voss, Stefan Fest, Ana Claudia Zenclussen

**Affiliations:** 1Department of Environmental Immunology, Helmholtz Center for Environmental Research, 04318 Leipzig, Germany; 2Department of Obstetrics and Gynecology, Städtisches Klinikum Dessau, Academic Hospital of University Brandenburg, 06847 Dessau-Rosslau, Germany; 3Department of Pediatrics, Städtisches Klinikum Dessau, Academic Hospital of University Brandenburg, 06847 Dessau-Rosslau, Germany; 4Saxonian Incubator for Translational Research, University of Leipzig, 04103 Leipzig, Germany

**Keywords:** 3D cell culture, placenta, hCG, spheroids, trophoblast migration, trophoblast invasion

## Abstract

It was postulated that 3D cell culture models more accurately reflect the complex tissue physiology and morphology in comparison to 2D cell monolayers. Currently, there is a shortage of well-characterized and easily maintainable high-throughput experimental models of the human placenta. Here, we characterized three different 3D cultures (e.g., spheroids) derived from trophoblast cell lines and studied their functionality in comparison to primary fetal trophoblasts and placental tissue. The spheroid growth rates of JEG3, BeWo and HTR8/SVneo cell lines were similar among each other and were significantly larger in comparison to primary trophoblast spheroids. All spheroids exhibited migratory properties and shortest distances were registered for JEG3 spheroids. Even though all spheroids displayed invasive capabilities, only the invasive features of HTR8/SVneo spheroids resulted in specific branching. This was in agreement with the invasive properties of the spheroids obtained from primary trophoblasts. Human chorionic gonadotropin production was highest in JEG3 spheroids and only increased when stimulated with cAMP and forskolin in BeWo, but not HTR8/SVneo spheroids. The gene expression analysis confirmed that 3D trophoblast cell cultures and especially HTR8/SVneo spheroids showed considerable similarities with the gene expression profile of primary placental tissue. This study offers a broad characterization of 3D trophoblast spheroids that, in turn, can help in selecting the best model depending on the scientific question that needs to be answered.

## 1. Introduction

Placentation is a complex multistep process comprised of trophoblast proliferation, survival, migration and invasion into the maternal decidua [1,2]. All of these processes are accompanied by hormone production and interaction with the local immune and stromal cells [3]. Adequate formation of the placenta ensures continuous nutrient, gasses and metabolite exchange with the fetus, along with limitation to xenobiotic and pathogen transfer [4,5]. Inadequate placentation can lead to pregnancy loss or development of pregnancy-associated disorders [6,7]. Currently, we still do not know the etiology of many pregnancy-associated disorders and as well as the placenta developmental processes are in a great need of further extensive investigation.

In order to improve current functional studies on human embryo implantation, defective placentation and reproductive and developmental toxicity, we need appropriate and practical in vitro models that more accurately portray in vivo placenta behavior. Several in vitro and in vivo models of placenta are already available to study placental functionality; however, there are still some inconsistences that need to be addressed. For example, in vivo models are mainly performed in rodents, which show different placenta morphology, mechanisms of migration and invasion and hormone production [8]. In vitro models consist mostly of culturing different types of immortalized trophoblast cell lines or primary cells grown in conventional two-dimensional (2D) conditions [9,10]. These 2D culture conditions have been proven to be poor predictors of treatment responses in vivo, mostly due to reduced cell-cell contact, flat layered cellular structure and increased surface area directly exposed to oxygen and nutrients [11,12]. More recently, the use of three-dimensional (3D) cell cultures, such as spheroids and organoids, have been proposed to more accurately resemble the tissue microenvironment [13,14]. Trophoblast stem cell-derived 3D cultures, e.g., organoids, represent the human morphology with high fidelity [15,16]; however, placenta tissue and isolation of trophoblast stem cells are not widely available to many scientists, with ethical and religious issues being a major impediment in many countries.

Three-dimensional spheroid cultures are generally single cell agglomerates that exhibit enhanced cell-to-cell contact and inter-cellular communication, with the outer segment composed of proliferative cells and an inner core that is filled with quiescent and necrotic cells [17]. Despite the fact that spheroids lack vascularization and cellular heterogeneity, it has been shown that gene expression profiles, pathway regulations and phenotype of 3D cultures resemble the ones observed in the tissue of origin [17,18,19]. Currently, spheroids have been successfully established from many tissues, including the ovary [20,21], liver [22], intestine [23], brain [24,25] and bone [26,27] and have been providing useful information regarding cellular differentiation, disease modeling and response to novel drug candidates. Although several studies report the usage of trophoblast spheroids [28,29,30,31], they are still largely under-represented in obstetrics preclinical studies as it is challenging for researchers to decide which model is the most appropriate to study particular placenta-related functions.

In order to understand the strengths and limitations of 3D cell cultures that can be employed for pregnancy-related preclinical studies, we characterized and evaluated 3D spheroid cultures of three main trophoblastic cell lines, i.e., BeWo, JEG-3 and HTR-8/SVneo, and study their functional characteristics in comparison to primary fetal trophoblasts (FTCs) and placental tissue.

## 2. Materials and Methods

### 2.1. Sampling of Primary Fetal Trophoblast Cells and Ethical Approval

Placental tissue was obtained from healthy women undergoing elective termination of pregnancy after informed consent was signed. The study and clinical protocol were approved by the ethics board at the medical association of Saxony-Anhalt, Halle, Germany (Ärztekammer Sachsen-Anhalt, Halle, Germany) with reference number 57/21. Placental villi were cleaned from blood cloths and scraped from the chorionic membrane. Next, the tissue was sequentially digested with 0.2% trypsin (Invitrogen, Karlsruhe, Germany) and 1 mg/mL collagenase V (Sigma-Aldrich, Steinheim, Germany). Cell suspensions were pooled and filtered through 100 µm nylon filters (Falcon, Durham, NC, USA). Red blood cells and debris were removed using density gradient centrifugation with a blood cell separation medium (MPbio, Eschwege, Germany). Cells were twice washed in PBS and grown in M-199 medium (Invitrogen, Germany) supplemented with 10% FBS (Biochrom, Berlin, Germany) and 100 µg/mL normocin (Invivogen, San Diego, CA, USA). Cultures were maintained in 5% CO_2_ in a humidified incubator at 37 °C. Cells were not used for more than 5 passages and were tested for mycoplasma contamination.

### 2.2. Cell Lines and Generation of Trophoblast Spheroids

The human choriocarcinoma JEG3 cell line was maintained in DMEM medium. BeWo cell line was grown in DMEM/F12 medium and HTR8/SVneo cell line in RPMI medium supplemented with 10 mmol/L HEPES, 100 nmol/L MEM nonessential amino acids and 1 mmol/L sodium pyruvate. Every medium was supplemented with 10% FBS and 1% penicillin-streptomycin (all media and reagents were from Invitrogen, Karlsruhe, Germany). For 2D cell cultures experiments, 5000 cells were plated per well in 96-well culture plates and analyzed within 24-72 h post seeding. For the establishment of the 3D spheroid formation, 500 to up to 3000 cells per well were plated in 96-well ultra-low attachment plates (Corning, Kennebunk, ME, USA). For the viability growth-rate assays, and the functional assays 1000 cells per well were plated for JEG3, BeWo and HTR8/SVneo cells and 3000 cells per well for primary fetal trophoblast cells in the above-mentioned media.

### 2.3. Cell Counting Assay

Live cell counting was performed using the Cell Counting WST-8 kit (Abcam, UK). In short, spheroids were collected at day 4, 7, 10 and 14 post-seeding. Then, spheroids were washed in PBS and dissociated using StemPro Accustase (Gibco, Grand Island, NY, USA) for 5 min at 37 °C and resuspended in 100 µL of corresponding medium. Afterwards, the dissociated single-cells were seeded in 96-well flat-bottom culture plates, and 10 µL WST-8 solution (Abcam, Cambridge, UK) was added directly to the cells. The cells were incubated at 37 °C for 4 h and the absorbance was measured at 450 nm using Infinite F200 microplate reader (Tecan, Grödig, Austria).

### 2.4. Viability Assay

Spheroids were plated in an ULA plate and viability was checked using multi-dye staining at day 1 and day 7 post-seeding. The staining was performed as previously described by Sirenko et al. [32]. In short, after the appropriate time of seeding, spheroids were stained without fixation with a mixture of dyes (Invitrogen, Eugene, OR, USA) in the following concentrations: 2 µM Calcein AM, 4 µM Ethidium homodimer and 33 µM Hoechst 33342. The dyes were added directly to the medium in order to minimize the disturbance of the spheroids. After 3 h of incubation, spheroids were imaged on fluorescent microscope (Keyence) using either 10× or 20× Pan Fluor objective. Fluorescence intensities were analyzed using ImageJ 1× software (National Institute of Health, Bethesda, MD, USA).

### 2.5. 3D Migration Assay

The migration assay was performed as previously described by Vinci et al. [33] with slight modifications. Cells were seeded at a density of 1000 cells/well in 96-well ULA plates. The medium was replaced on the 4th day and afterwards, the spheroids were transferred to 0.1% gelatin- (Sigma-Aldrich, Darmstadt, Germany) coated 96 well-flat bottom cell culture plates and were allowed to adhere to the bottom (one spheroid per well in 300 µL medium). Images were taken using a Keyence BZ-X800 microscope at 24, 48 and 72 h post transfer. Images were analyzed using BZ-X800 Analyzer software. For each spheroid the total area (spheroid plus outgrown migration area) and the migration area (outgrown area excluding the central spheroid part) were manually determined as the visible border of the outer rim (spheroid) and the outer cells of the migrating front. Average radius of migration was carried out from the outer rim of the spheroid. For each time point at least five spheroids were analyzed.

### 2.6. 3D Invasion Assay

Cells were seeded in ULA plates and after 96 h the complete medium was carefully removed from the wells. Then, 200 µL of 2,5 mg/mL Matrigel Basement Membrane Matrix, LDEV-free (Corning, Bedford, MA, USA) dissolved in complete medium (depending on the cell line used) was added on the side of each well and allowed to solidify for 15 min at 37 °C. Afterwards, 100 µL of complete medium was added and the images of the spheroids were taken at 0, 24, 48 and 72 h post Matrigel embedding using a Keyence BZ-X800 microscope.

### 2.7. β-HCG Quantification

Cell supernatants from 2D and 3D trophoblast cell lines were collected after 48 h in culture, with or without 50 µM cAMP (Sigma-Aldrich, St. Louis, MO, USA) and 10 µM forskolin (Sigma-Aldrich, Darmstadt, Germany) stimulation. All supernatants were centrifuged for 15 min at 14000 rpm at 4 °C using a 3K Amicron ultra-centrifugal filter (Merck, Tullagreen, Co., Cork, Ireland) and stored at −80 °C until further analysis. β-HCG quantification was obtained using human β-HCG ELISA detection assays (DRG, Marburg, Germany) according to manufacturer’s instruction. Absorbance was measured using an Infinite F200 microplate reader (Tecan, Grödig, Austria).

### 2.8. RT-PCR

Total RNA was isolated from 8 to 12 spheroids using TRIzol (Invitrogen, Carlsbad, CA, USA), and RNA quantity and quality were checked using Infinite F200 Nanoquant (Tecan, Grödig, Austria). Total RNA (600 ng) was used for cDNA synthesis and RT-PCR procedure, and analysis was performed as previously reported [7].

### 2.9. Statistical Analysis

All assays were repeated three times (*n* = 3), and in every assay the samples were analyzed in triplicate unless stated otherwise. Experimental results are presented as mean ± SD. The data statistical analysis was performed using GraphPad Prism software version 8.0 (GraphPad Software, San Diego, CA, USA) using one- or two-way ANOVA followed up by Tukey’s multiple comparison test. Statistically significant differences were regarded when *p*-value was less than 0.05.

## 3. Results

### 3.1. Growth Rate Follow-Up of 3D Trophoblast Cell Lines

In order to compare the growth differences between 3D cultures of BeWo, JEG3, HTR8/SVneo and primary trophoblast cells (FTCs), comparable number of cells were plated per well in ultra-low attachment (ULA) plates. The number of cells used per well and per cell line was chosen on the basis of previous experiments that determined the diameter of spheroids after 24 h culturing in ULA plates (Appendix A). For the trophoblast cell lines, we used 1000 cells/well (Appendix A) and for the primary FTCs we used 3000 cells/well (Appendix A). Pictures of the spheroids were taken every day (24 h post seeding) for up to two weeks. As shown in Figure 1A, BeWo, JEG3 and HTR8/SVneo spheroids showed similar configuration with a round outline, while FTC spheroids showed an irregular shape and uneven edges after 24 h in culture. After 48 h they formed small and sold circular globules that maintained their shape for the following two weeks (Figure 1A). The number of live cells per spheroids was quantified at day 4, 7, 10 and 14 post seeding and FTC spheroids contained significantly less metabolically active cells in comparison to the rest of the spheroids (Figure 1B). Furthermore, the analysis of the total spheroid area (perimeter) and mean spheroid diameter confirmed that FTC spheroids remained significantly smaller (~50 µm), while BeWo, JEG3 and HTR8/SVneo spheroids reached a diameter of ~300–400 µm within two weeks (Figure 1C,D).

### 3.2. Viability Evaluation of 3D Trophoblast Cell Lines

As spheroids tend to have a necrotic core due to the limited oxygen availability in the center of the spheroids [34], we next addressed whether the trophoblast spheroid viability was affected over prolonged periods of culturing in the 3D setup. We performed a one-step staining with three different dyes in order to stain the nuclei (with Hoechst); the metabolically active (with Calcein AM) and the necrotic cells (with Ethidium Dimer) at day 1 (Figure 1A) and day 7 post-seeding (Figure 2B). The fluorescence intensity of metabolically active cells at day 1 and day 7 was similar among the different trophoblast spheroids, although FTC spheroids showed significantly lower fluorescence intensity in comparison to HTR8/SVneo at 7 days post-seeding (Figure 2C). The fluorescence intensity of the necrotic cells was significantly higher in FTC spheroids at day 1 (Figure 2D), while after 7 days in culture almost all 3D spheroids showed increased fluorescence intensity of necrotic cells (Figure 2D), indicating decreased viability after one-week perseverance in culture.

### 3.3. Migratory Properties of 3D Trophoblast Cell Lines

In order to evaluate the migratory potential of different trophoblast spheroids, we transferred them from ULA plates to gelatin-coated flat bottom 96-well culture plates and observed the migratory area over a period of 72 h. All spheroids showed migratory advantage (Figure 3A); however, HTR8/SVneo and BeWo spheroids showed the largest migratory areas and migratory distances (Figure 3B–D) in comparison to JEG3 spheroids. Only in the first 24 h, BeWo spheroids showed greater migratory areas when compared to HTR8/SVneo spheroids (Figure 3B,C).

### 3.4. Invasion Rates of 3D Trophoblast Cell Lines

A crucial aspect of placentation is the ability of the trophoblasts to invade into the remodeled endometrial stroma following the basement membrane and extracellular matrix. In order to investigate and compare the invasive capacities of different 3D trophoblast spheroids we cultured them into a gel-like extracellular matrix, e.g., Matrigel. The overall Matrigel concentration used was 2.25 mg/mL to allow spontaneous invasion. The invasive protrusions or invadopodia were monitored at intervals within 72 h and all trophoblast spheroids already showed invasive capacities at 24 h post-seeding (Figure 4A). While BeWo and JEG3 spheroids showed budding protrusions in all directions into the Matrigel, only HTR8/SVneo spheroids showed uniform radial branching into the extracellular matrix which was comparable to the invasive capacities of FTC spheroids (Figure 4A). Although the total invasive area was comparable among the spheroids at different time points (Figure 4B), the HTR8/SVneo and FTC spheroids showed similar invasive rates from the center that were significantly greater in comparison to the invasive distances of JEG3 and BeWo spheroids.

### 3.5. Growth Rate Follow-Up of 3D Trophoblast Cell Lines

To assess the secretion of human chorionic gonadotropin (hCG), cell supernatants of 2D and 3D trophoblast cell lines were analyzed for the beta subunit of hCG 96 h post seeding (Figure 5A). Interestingly, JEG3 cells cultured in a conventional 2D way yielded the highest β-HCG concentrations (318,6 ± 27.49 mIU/mL) in comparison to BeWo (4.5 ± 0.6 mIU/mL) and HTR8/SVneo cells (13.01 ± 9.5 mIU/mL). In contrast, 3D cell culture led to a further increase in β-HCG production in BeWo and JEG3 cells to 287 ± 144.1 mIU/mL and 787.0 ± 142.4 mIU/mL, respectively. HTR8/SVneo cell lines did not secrete any additional β-HCG (10.5 ± 1.1 mIU/mL) in comparison to 2D HTR8/SVneo (Figure 5A).

To further investigate whether hCG production can be amplified in BeWo and HTR8/SVneo cells, we stimulated these 3D cell lines with forskolin or cAMP for a further 48 h. Forskolin and cAMP are known inducers of morphological or biochemical trophoblast differentiation and can lead to increased β-HCG secretion [35]. As expected, 48 h stimulation with either 50 µM cAMP or 10 µM forskolin lead to a 5-fold increase in β-HCG production in BeWo 3D culture (Figure 5B). On the contrary, no β-HCG secretion was detected in HTR8/SVneo spheroids (Figure 5B).

### 3.6. Comparison of Trophoblast Specific Gene Expression in 3D Trophoblast Spheroids

Next, we wanted to investigate the transcriptome-level reliability of 2D and 3D cultures of trophoblast cell lines as compared to first trimester placenta tissue. We used gene expression analysis to verify the level of expression of a series of well-established genes important for placental development and function. As expected, placental tissue had a distinct gene expression pattern, and 2D trophoblast cell cultures showed a number of significantly different expressed genes, including the cytochrome p450 CYP1B1, cell adhesion markers CDH2 and AKR1C1, cell invasion markers MMP9, LGALS3, FOSL1 and the transcriptional factor STAT3 (Figure 6A). On the contrary, 3D cultures accurately resembled more the gene expression patterns of the placental tissue, and specifically HTR8/SVneo spheroids, showed the least significant differences with first trimester placental tissue (Figure 6B).

Finally, we performed a time-course study and assessed the gene expression profile of trophoblast spheroids at different time points. Expression analysis of CYP1B1—an enzyme important for metabolism of steroid hormones—showed that it is highly expressed in the first 4 to 7 days post-seeding and decreases at day 14 (Figure 7A–C) in all trophoblast spheroids. Similar expression patterns were observed for the cell adhesion and invasion markers AKR1C1 and MMP9 (Figure 7D–I). The multipotent transmembrane receptor ITGB1 showed relatively high expression also on day 7 post seeding in all spheroids (Figure 7J–L). The least differences over time were recorded for STAT3 transcriptional factor (Figure 7M–O). In summary, after 10 to 14 days in culture, trophoblast spheroids decreased their gene expression levels for most of the genes important for studying placenta functionality.

## 4. Discussion

With the current study, we report for the first time a comparative evaluation of different trophoblast spheroids on the following functional parameters: (A) proliferation, (B) viability, (C) migration, (D) invasion, (E) hormone production and (F) molecular traits similarities with human placental tissue. As there is an increasing demand for more cost-effective 3D models of placenta and since animal experimentation is costly and morally demanding, these well-characterized 3D spheroids could be utilized for modeling pregnancy complications or toxicology testing.

One key parameter for 3D culture establishment is the assessment of spheroid growth and viability over certain periods of time. Due to the 3D assembly of the spheroids and the absence of a supportive scaffold, the availability of sufficient nutrients, oxygen and pH parameters are not the same throughout the spheroid [36] and might lead to decreased proliferation rates. We showed that prolonged 3D culturing and recurring medium replacement lead to a steady and increasing spheroid growth for BeWo, JEG3 and HTR8/SVneo spheroids, while the growth rate was rather slow and diminishing for primary trophoblast spheroids in the same timeline. This agrees with previous studies that have shown that even if optimal growth conditions are provided and initial cell densities are increased, primary cells even in a 3D culture setup have limited growth potential [37,38,39]. Although there is a possibility for medium adjustment and inclusion of additional growth factors in the medium in order to preserve the stemness and proliferative state of primary trophoblast cells [40], in this study we have not explored this and it might be the focus of subsequent studies. Importantly, as all trophoblast spheroids showed a great proportion of viable cells even after 7 days in culture, this period would be optimal for cytotoxic effect evaluation of drug candidates or environmental toxins in future studies.

Initial migration and consecutive invasion of trophoblasts are crucial processes in placentation. These two properties are of special relevance as their failures in terms of slow trophoblast migration and shallow trophoblast invasion can lead to pregnancy-associated disorders, such as pre-eclampsia, fetal growth restriction or spontaneous abortion [7,41]. Even though it has been reported that all of the above-used trophoblast cell lines show migratory properties in a 2D monolayer [9], we unequivocally show here that JEG3 spheroids display the lowest migration rate in the surrounding area. This might be explained by the different levels of proteases and cell adhesion molecules [9]. Previously, a comparative study of HTR8/SVneo and JEG3 2D cultures reported that HTR8/SVneo cells show increased invasiveness into the extracellular matrix which was due to the differentially expressed proteases and inhibitors of proteases [42]. In our study, we observed that BeWo and JEG3 spheroids invade the surrounding extracellular matrix differently than spheroids derived from primary trophoblast cells. This is in line with the results reported by Oraiopoulou et al. [25] that primary spheroids (in their case, glioblastoma) have a distinctive invasive pattern that contrasts the invasion of spheroids derived from conventional glioblastoma cell lines. Remarkably, we observed that HTR8/SVneo spheroids show almost identical extensive branching invasion as the primary spheroids and do not recapitulate the budding-like expansion of the rest of the spheroids. Taking this into account, HTR8/SVneo spheroids may be a useful model in exploring the physiological and molecular patterns of trophoblast invasion.

Several studies have shown that 2D and 3D cultures differ in their gene expression profiles, especially for the genes involved in cell survival, proliferation, migration and invasion [29,37,43,44,45]. This was also the case in our study and could reflect the specific composition of the 3D spheroids which contain proliferative, quiescent, apoptotic and necrotic cells. These cell differentiation processes are also regularly present during the process of placentation [46]. Furthermore, HTR8/SVneo spheroids showed the least differentially expressed genes involved in cell proliferation, survival and invasion when compared to first trimester placental tissue. This can be explained by the fact that HTR8/SVneo cell lines are derived from first trimester extravillous trophoblasts that has been immortalized by transfection of the gene encoding simian virus large T antigen [47]. This shows that, in addition to the functional properties of cell invasion shared between 3D spheroids of HTR8/SVneo cells and primary trophoblast cells, HTR8/SVneo spheroids also resembles the gene expression traits of the placental tissue.

Furthermore, we observed similar kinetics of the genes important for placenta functionality in all spheroids. There was matched and time-dependent expression of CYP1B1, AKR1C1, MMP9 and ITGB1 genes important for hormone metabolism, migration and invasion in the placenta. Consecutive decreases in these markers after 10 to 14 days in culture can be expected due to the increase in the necrotic core, limited nutrient availability, motility or the mechanical integrity of the aged spheroids [48,49]. However, the gene expression of STAT3 transcriptional factor was not changed over the same time, which was expected due to its role in rapid tumor growth and inflammation-driven tumorigenesis [50,51]. Together, our gene expression kinetics analyses showed that trophoblast spheroids at day 4 and day 7 post-seeding share the gene expression profiles and that both time points are suitable for screening platforms.

As the placenta is an endocrine organ, hormone production and release should be an important aspect of a 3D trophoblast cell model. hCG is one of the first hormones secreted by the trophoblasts and the highest hCG values are recorded by the end of the first trimester of pregnancy [52]. Furthermore, it was postulated that it has a critical role in embryo implantation and placentation [53] and in immune cell modulation [3]. In 2D cultures only JEG3 cells secreted substantial amounts of hCG in the cell supernatant; however, in the 3D cell culture setup not only JEG3, but as well BeWo spheroids showed facilitated hCG secretion. On the contrary, this was not the case for HTR8/SVneo spheroids. These differences were not due to the initial cell number, spheroid growth and viability as previously shown in Figure 1, meaning that the hormone production is influenced by factors specific to the cell type and cell origin. It was recently shown that HTR8/SVneo cells show a spontaneous transition from epithelial to mesenchymal phenotype in order to promote invasion of the extravillous trophoblast in the surrounding maternal endometrium [54]. hCG is primarily secreted from the syncytiotrophoblast which shows an epithelial phenotype, as well as from JEG3 and BeWo cells, which could be an explanation as to why we do not see any hCG secretion in HTR8/SVneo spheroids. This could not have been improved even with the addition of cAMP or forskolin [55], which are known inducers of syncytialization, by activation of protein kinase A and induction of trophoblast fusion.

## 5. Conclusions

The present study is the first to report detailed evaluation and comparison of 3D trophoblast spheroids. It demonstrates that HTR8/SVneo spheroids are a potential, highly scalable and easily maintainable spheroid model to study placental proliferation, viability, migration, invasion and molecular traits. Nevertheless, when addressing research questions regarding the endocrine placenta function, JEG3 and BeWo spheroids are more compatible model. We are confident that the results obtained in the present study are of great interest for scientists when evaluating which system is the best-suited for their particular placenta-related scientific question.

## Figures and Tables

**Figure 1 cells-11-02884-f001:**
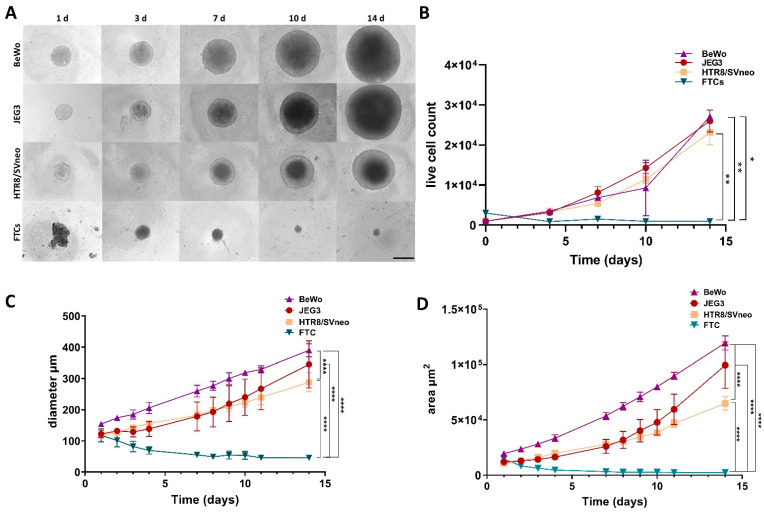
Trophoblast spheroid growth curve. Bright-field images of BeWo, JEG3, HTR8/SVneo and FTC spheroids at day 1, 3, 7, 10 and 14 (**A**), number of live cells per spheroid at day 4, 7, 10 and 14 (**B**) and growth curves for the representative trophoblast spheroids over a period of two weeks as a function of total spheroid area in µm^2^ (**C**) or spheroid diameter in µm (**D**). Scale bar 100 µm. * *p* < 0.05; ** *p* < 0.01; **** *p* < 0.0001.

**Figure 2 cells-11-02884-f002:**
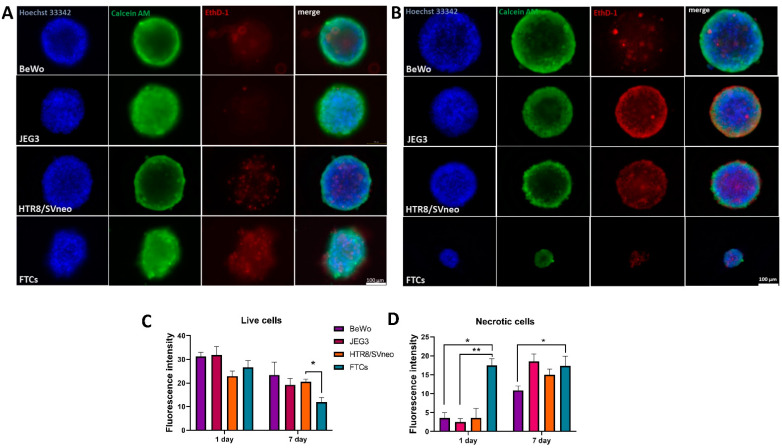
Viability prevalence in trophoblast spheroids. Fluorescent images at day 1 (**A**) and day 7 (**B**) showing metabolically active cells with Calcein AM staining (green) and necrotic cells with Ethidium Dimer (red). Nuclei are stained with Hoechst 33342 (blue). Fluorescence intensities of live and metabolically active (**C**) and necrotic (**D**) cells. Scale bar 100 µm. * *p* < 0.05; ** *p* < 0.01.

**Figure 3 cells-11-02884-f003:**
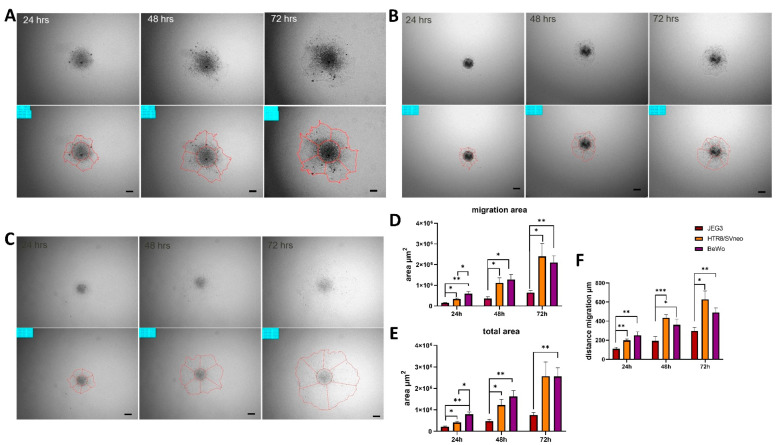
Comparison of migration in trophoblast spheroids. Bright-field images of BeWo (**A**), JEG3 (**B**) and HTR8/SVneo (**C**) at 24, 48 and 72 h and lower panels show the contours that were manually segmented. Comparison of total migratory area (**D**), total spheroid area (**E**) and migratory distances (**F**). Scale bar 100 µm.* *p* < 0.05; ** *p* < 0.01; *** *p* < 0.001.

**Figure 4 cells-11-02884-f004:**
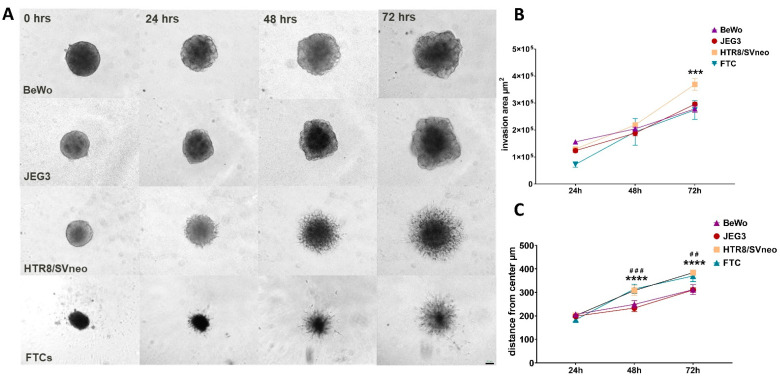
Invasive properties of trophoblast spheroids. Bright-field images of BeWo, JEG3, HTR8/SVneo and FTCs spheroids at 0, 24, 48 and 72 h; scale bare 100 µm (**A**). Comparison of total invasion area in µm^2^ (**B**) and distance of invasion starting from the center in µm (**C**). *** *p* < 0.001; **** *p* < 0.0001 versus HTR8/SVneo spheroids and ## *p* < 0.01; ### *p* < 0.001 versus FTCs spheroids within the same time point.

**Figure 5 cells-11-02884-f005:**
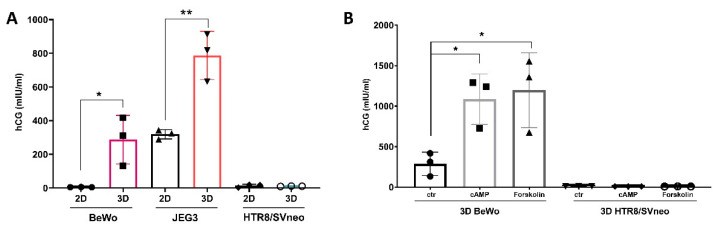
Human chorionic gonadotropin production in trophoblast spheroids. Comparison of hCG production in 2D vs. 3D trophoblast cell cultures (**A**) and after cAMP and forskolin stimulation (**B**). * *p* < 0.05; ** *p* < 0.01.

**Figure 6 cells-11-02884-f006:**
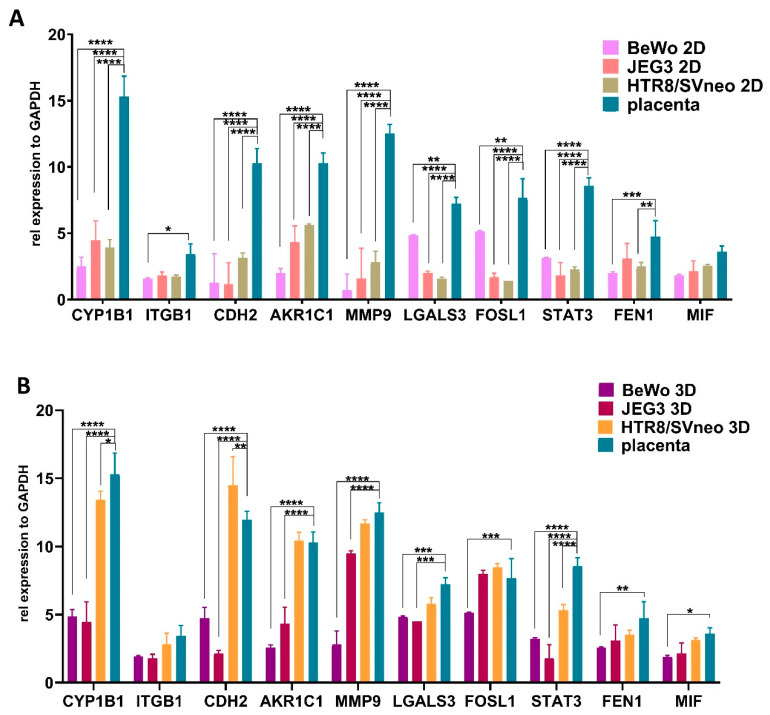
Gene expression profiles in trophoblast spheroids. Comparison of gene expression levels of cytochrome p450 gene CYP1B1; cell adhesion markers CDH2 and AKR1C1; cell invasion markers MMP9, LGALS3, FOSL1 and MIF; DNA damaging factor FEN1 and the transcriptional factor STAT3 between 2D cell cultures of BeWo, JEG3 and HTR8/SVneo cell lines (**A**) and respective 3D spheroids (**B**) with first trimester placenta tissue. * *p* < 0.05; ** *p* < 0.01; *** *p* < 0.001; **** *p* < 0.0001.

**Figure 7 cells-11-02884-f007:**
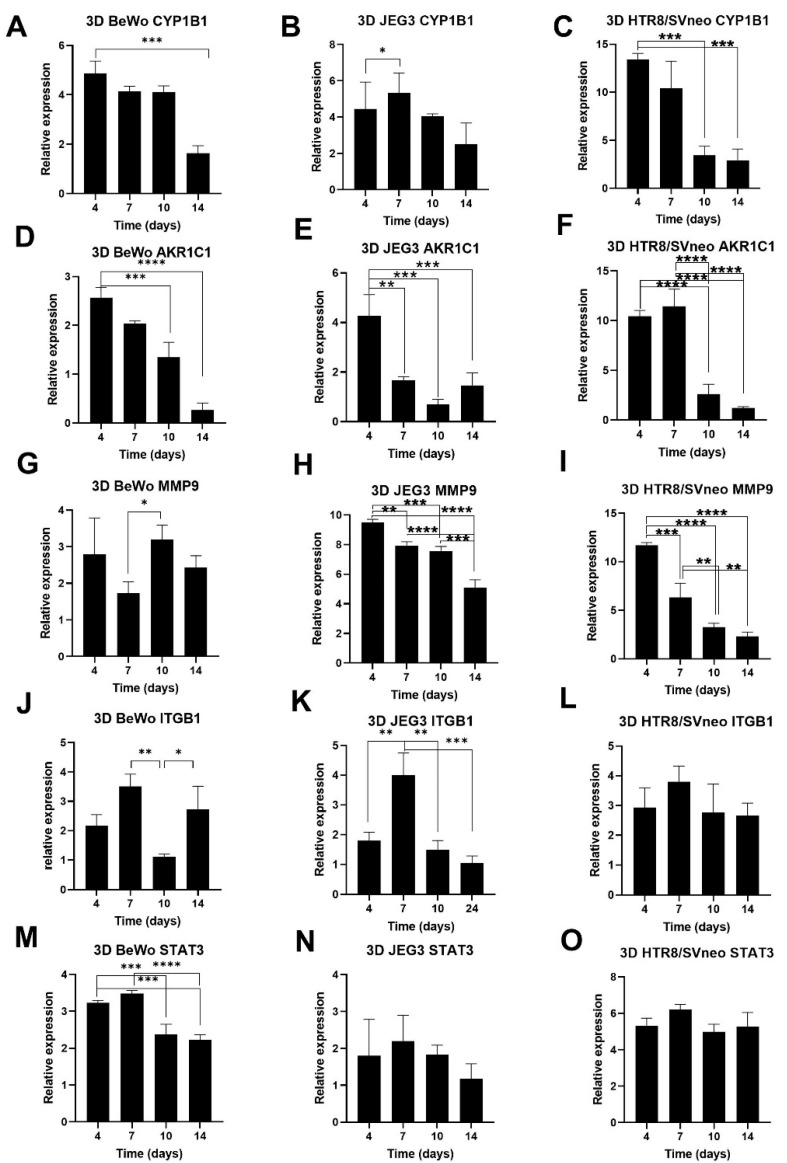
Gene expression kinetics in trophoblast spheroids. Time-course gene expression was performed in BeWo, JEG3 and HTR8/SVneo spheroids for CYP1B1 (**A**–**C**), AKR1C1 (**D**–**F**), MMP9 (**G**–**I**), ITGB1 (**J**–**L**) and STAT3 (**M**–**O**) at day 4, 7, 10 and 14 post-seeding. * *p* < 0.05; ** *p* < 0.01; *** *p* < 0.001; **** *p* < 0.0001.

## Data Availability

Not applicable.

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
