# Peer review of "Characterization of Three-Dimensional Trophoblast Spheroids: An Alternative Model to Study the Physiological Properties of the Placental Unit"

_cells, 2022, doi:10.3390/cells11182884_

Round 1

Reviewer 1 Report (Previous Reviewer 1)

The paper is now improved but the figures especially 1.2 and 3 needs better resolution with 300 dpi please try to use more recent references. 

Reviewer 2 Report (Previous Reviewer 3)

.

This manuscript is a resubmission of an earlier submission. The following is a list of the peer review reports and author responses from that submission.

Round 1

Reviewer 1 Report

The study is of good merit as paragraphs are written well but some recommendations are needed:

the aim must be rephrased again , ethical approval number  should be provided in the materials section addtion  the discussion  section  should be summarized to be more accurate to the  readers line 120 -126 need paraphrasing   line 170-190 need to be rewritten  again.......please try  to provide more relevant  references please  also provide figure with high quality and better resolution 

Author Response

We thank the reviewer for the nice recommendations. Now we have rephrased the aim and is replaced with (Page 1, Line 43-45): In order to improve current functional studies on human embryo implantation, defective placentation and reproductive and developmental toxicity, we need appropriate in vitro models that more faithfully embrace the in vivo placenta behavior.

The ethical approval number is also mentioned in the materials and methods section and additionally at the end of the paper (Page: 2, Line: 83 and Page 12, Line: 394).

We have added a conclusion section additionally in order to summarize it shortly the discussion part. This can be found on page 11, line 367-375.

Lines 120-126 are now phrased into:3D Invasion assay. Cells were seed in ULA plate and after 96 hours the complete medium was removed carefully from the wells. 200 µl of 2,5 mg/ml Matrigel Basement Membrane Matrix, LDEV-free (Corning, Bedford, MA, USA) dissolved in complete medium (depending on the cell line used) was added on the side of each well, and allowed to solidify for 15 mins at 37°C. Afterwards 100 µl of complete medium was added and the images of the spheroids were taken at 0, 24, 48 and 72 hours post Matrigel embedding at Keyence BZ-X800 microscope (Page 3, lines 130-136). 

and for the cell viability paragraph at lines 170-190 we have now included the fluorescent intensity quantification to the figure in order to increase the comprehension of the paragraph (Page 4, lines 181-190).

We have updated the references list, as suggested by reviewers.

Due to the final formatting into the Cells journal format, some of the figures indeed seem small, and we apologize for any inconveniences. Now we have additionally submitted the original 300 dpi figures together with the revised manuscript and we believe that this will increased the quality and resolution necessary for publication.

Reviewer 2 Report

7th July, 2022

Review of the Manuscript ID: cells-1824799, by V. Stojanovska et al., entitled: “Characterization of three-dimensional trophoblast spheroids: an alternative model to study the physiological properties of the placental unit” that is intended to be published as the Article in Cells

(separate Microsoft Word file as Reviewer Attachment for Manuscript ID cells-1824799 Cells 7th July 2022 that includes Comments to the Authors is also uploaded)

Taking into account research highlight, contribution of the Authors to the progress in the research field, thorough manner of data presentation, perfectly writing in English, abundance of Materials and Methods, Results as well as diligent graphic documentation and photographic visualization, the quality of this paper deserves praise and merits my support. The Authors have received the high scores from me for the originality, importance of the work and the scientific value of their paper. In my opinion, the current paper provides insightful interpretation of topical and coming trends in the developing and optimizing the 3D model of ex vivo expansion of trophoblast spheroids in order to comprehensively explore the cytophysiological functions, molecular attributes and transcriptomic signature of human extravillous trophoblast-related placental compartment. Collectively, considering all these facts, I strongly recommend the Editorial Board to allow for publication of this very interesting paper in Cells, after the minor revision of the manuscript will have been completed by the Authors and provided that the Authors are ready to consider all the Reviewer comments indicated below:

1) There is a lack of the separate Conclusions section in the paper. Therefore, this section should have been added at the end of the manuscript to thoroughly summarize the manuscript and include important paragraphs targeted at presentation of research highlight and future research directions.

2) There is a lack of the separate Abbreviations section in the paper. Therefore, this section should have been added to thoroughly elucidate and expand a broad spectrum of the in-text abbreviations, which have been used by the Authors in all the sections of their paper.

3) The References section has to be prepared in the format compatible with the requirements of Cells.

General Comment of the Reviewer:

Before the manuscript will have been accepted for publication in Cells, it requires the minor revision (according to all the remarks and suggestions of the Reviewer).

Author Response

  • There is a lack of the separate Conclusions section in the paper. Therefore, this section should have been added at the end of the manuscript to thoroughly summarize the manuscript and include important paragraphs targeted at presentation of research highlight and future research directions.

We thank the reviewer for the nice comments and recommendations. We completely agree that a conclusion section is necessary and we have added one at page 11, line 367-375.

  • There is a lack of the separate Abbreviations section in the paper. Therefore, this section should have been added to thoroughly elucidate and expand a broad spectrum of the in-text abbreviations, which have been used by the Authors in all the sections of their paper.

Agreed and added to the final manuscript (Page 12, Line 378-385)

  • The References section has to be prepared in the format compatible with the requirements ofCells.

We sincerely apologize for this inconvenience and we have updated the references format to the Cells journal requirement.

Reviewer 3 Report

The paper is interesting and valuable but some description or discussion is poor. The readers must be confusing without some sentences. At the current version, the reviewer cannot recommend the publication. Taken together, major revisions should be made before re-submission. The paper would be re-considered only when all the comments were responded.

1. Lines 54-66.

These parts are explanations of 3D spheroids. This is the most important concept in this manuscript. However, the contents are too poor to understand the study clearly.

The authors should introduce the concept of 3D spheroids applied to several representative tissues. Are there little papers on the placenta by using 3D spheroids? The authors should reveal the points. At the current version, the reviewer cannot recommend the publication.

To reduce the authors’ burden, I suggest the sentences or references (both review and research paper should be quoted.) to be added for the revision.

3D spheroid models, such as the nerve [], cancer [], bone [], or cartilage [], have been recently noted to evaluate cell ability or therapeutic efficiency. These systems enabled the tissue microenvironment and promotion of the development of new drug candidates or novel therapeutic effect. However, there are few papers on placental……….

Review and research paper

Nerve

doi.org/10.1016/j.biomaterials.2017.10.002

10.1039/D0LC01112F 

Cancer

Cancers 202012(10), 2754

Tissue Eng. Part C Methods 201925, 711–720. https://doi.org/10.1089/ten.tec.2019.0189

Tissue Eng. Part A, 26, 2020, 1272-1282. https://doi.org/10.1089/ten.tea.2020.0095

Bone

doi.org/10.1016/j.phrs.2021.105626

doi.org/10.1016/j.biomaterials.2020.120607

doi.org/10.1016/j.bone.2019.05.018

Cartilage

org/10.1089/ten.teb.2020.0354

doi.org/10.1002/advs.202103320

2. Viability assay and results

The authors should perform not only the live/dead staining but also quantitive evaluation using such as WST-8 kit. The viability evaluated by statistical analysis is essential.

3.

The time course of cell number per spheroid should be investigated.

4. Figure 3

The scale bar is too small to recognize.

5. Figure 6

The time course of gene expression should be evaluated. The kinetics are important.

6. Discussion
This part can not be understood. The authors should discuss the strength by quoting related papers above. Unfortunately, there are few references quoted.

Author Response

  1. Lines 54-66...

We thank the review for the nice recommendations and for pointing out the gap of the knowledge in the field of trophoblast spheroids. So, we have adjusted this to the text and included some and others relevant references as pointed out by the reviewers. Now this can be tracked at page 2, lines 65-72.

  1. Viability assay and results

The authors should perform not only the live/dead staining but also quantitive evaluation using such as WST-8 kit. The viability evaluated by statistical analysis is essential.

We thank the reviewer for this comment and we fully agree that quantitative evaluation is necessary and we have performed that in ImageJ and have included in the figure and in the materials and methods section. This just confirmed, what we have previously observed macroscopically. Unfortunately, we did not perform another live/dead staining, firstly due to time constrains and secondly because we want to promote this one-step staining method as a fast and reliable that without additional manipulation of the spheroids (and addition of extra errors) we can easily track the metabolically active vs necrotic cells (aka live/dead).

  1. The time course of cell number per spheroid should be investigated.

We understand the point of the reviewer and this is something that we are exploring in a next set of reproductive toxicology studies. In this study we wanted to describe the state of the art of trophoblast spheroids growth and expansion and for that cause we have followed their diameter and total area growth area in a period of two weeks as a dependent variables to model in future the growth of the spheroids. We hope that the reviewer will agree on this with us.

  1. Figure 3: The scale bar is too small to recognize.

We agree with the reviewer and we have adjusted the figure for better visibility and readability.

  1. Figure 6: The time course of gene expression should be evaluated. The kinetics are important.

We agree with the reviewer, and as we mentioned earlier before, kinetics is something that we would focus on in another set of pregnancy toxicology studies. With this manuscript we aimed primarily to compare the available trophoblast spheroids and show a set of validated techniques to easy access their functionality for future studies.  

  1. Discussion

This part can not be understood. The authors should discuss the strength by quoting related papers above. Unfortunately, there are few references quoted.

We agree with the reviewer and in that order, we have reinforced our discussion part with some of the previously mentioned references and now all the changes can be tracked in red at pages 10 and 11; lines 322-340.

Round 2

Reviewer 3 Report

  1. We understand the point of the reviewer and this is something that we are exploring in a next set of reproductive toxicology studies. In this study we wanted to describe the state of the art of trophoblast spheroids growth and expansion and for that cause we have followed their diameter and total area growth area in a period of two weeks as a dependent variables to model in future the growth of the spheroids. We hope that the reviewer will agree on this with us.

To claim growth, cell number is the essential factor. The reviewer can not agree with the authors' comments. The diameter of spheroids is large so cells may lose the appropriate function. Taken together, cell number kinetics are essential. To publish the Cells (IF:7.6), this data is needed. 

  1. We agree with the reviewer, and as we mentioned earlier before, kinetics is something that we would focus on in another set of pregnancy toxicology studies. With this manuscript we aimed primarily to compare the available trophoblast spheroids and show a set of validated techniques to easy access their functionality for future studies.

The diameter of spheroids is large so cells may lose the appropriate function. To claim the function, the authors must perform the experiment.

The reviewer can not recommend the publication because the authors cannot reply to the appropriate and logical sentences.

That's all.

Author Response

2nd Response to Reviewer 3:

Concern 1 

To claim growth, cell number is the essential factor. The reviewer can not agree with the authors' comments. The diameter of spheroids is large so cells may lose the appropriate function. Taken together, cell number kinetics are essential. To publish the Cells (IF:7.6), this data is needed. 

Answer 1: We would like to thank the reviewer for this observation and we took the time to perform additional experiments and implement it to the manuscript. We have followed the growth rate of the trophoblast spheroids over a period of 2 weeks and we assessed the spheroid cell number via the WST-8/CCK8 cell counting kit as suggested by the reviewer at the following time points: day 4, day 7, day 10 and day 14. Now these data are included in Figure 1B (page 5) and lines 174-177, page 4 of the manuscript.

Concern 2

The diameter of spheroids is large so cells may lose the appropriate function. To claim the function, the authors must perform the experiment.

Answer 2: Indeed, we agree with the reviewers concern that gene expression kinetics are important in following up the spheroid’s functionality and now we have added these experiments to our manuscript. We followed up the expression of the following genes: CYP1B1 (important for steroid hormone metabolism), AKR1C1 and ITGB1 (important for cell migration and adhesion), MMP9 (important for trophoblast invasion) and STAT3 (transcriptional factor important for tumor growth and development), again at day 4, day 7, day 10 and day 14. We observed that for most of the genes the expression is stable in the first two time-points and then it declines by day 10 and day 14, which indicates possible disturbed functionality of the spheroids after prolonged periods 3D cell culture. These data are now implemented in the original manuscript in additional Figure 7 (page 10) and lines 278-288, page 10 and lines 354-364, page 12.

We hope that the reviewer agrees with us that with the addition of these extra experiments as kindly suggested by the reviewer has improved our manuscript substantially.